

# The Spanish version of the Fatigue Assessment Scale: reliability and validity assessment in postpartum women

Antoni Cano-Climent[1], Antonio Oliver-Roig[2,*], Julio Cabrero-García[2,*], Jolanda de Vries[3] and Miguel Richart-Martínez[2,*]

[1] Hospital General d'Ontinyent, Conselleria de Sanitat Universal i Salut Pública, Ontinyent, Valencia, Spain
[2] Department of Nursing, University of Alicante, San Vicent del Raspeig, Alicante, Spain
[3] CoRPS Center of Research on Psychology in Somatic Diseases, Department of Medical and Clinical Psychology, University of Tilburg, Tilburg, Netherlands
[*] These authors contributed equally to this work.

## ABSTRACT

**Background**. Fatigue is the most widely reported symptom by women during pregnancy, labour, the postpartum period, and early parenting. The objective was to translate the Fatigue Assessment Scale (FAS) into Spanish and assess its psychometric properties.

**Methods**. Instrumental Design. The FAS was translated into Spanish (FAS-e) using forward and back translation. A convenience sample was constituted with 870 postpartum women recruited at discharge from 17 public hospitals in Eastern Spain. Data was obtained from clinical records and self-administered questionnaires at discharge. Internal consistency, factor structure, comparisons between known groups and correlations with other variables were assessed.

**Results**. Cronbach's alpha coefficient was .80. Findings on the dimensionality of the FAS-e scale indicated that it was sufficiently unidimensional. FAS-e scores were higher among women who had undergone caesarean births ($p < .05$), had a higher level of postpartum pain ($p < .01$), experienced difficulties during breastfeeding ($p < .01$) and had lower levels of self-efficacy for breastfeeding ($p < .01$).

**Conclusions**. An equivalent Spanish version of the FAS was obtained with good reliability and validity properties. FAS-e is an appropriate tool to measure postpartum fatigue.

Corresponding author
Antonio Oliver-Roig,
Antonio.Oliver@ua.es

## INTRODUCTION

Fatigue is a complex, multi-causal, multi-dimensional, non-specific, and subjective phenomenon that has no widely accepted single definition (*Tiesinga, Dassen & Halfens, 1998*). Fatigue appears when demands of a process or situation exceed available resources and recovery mechanisms are not sufficient (*Aaronson et al., 1999*). One attempt to standardize the definition of fatigue can be found in the Medical Subject Headings (MeSH) controlled vocabulary, where fatigue is defined as "the state of weariness following a period of exertion, mental or physical, characterized by a decreased capacity for work and reduced efficiency to respond to stimuli" (*National Library of Medicine, 2016*).

Fatigue during pregnancy, labour, the postpartum period, and early parenthood is a common experience. Its prevalence is highly variable, ranging between 15% and 76% (*Cheng & Li, 2008*), depending on the measurement instrument, culture, and period measured (*Cheng et al., 2014*).

Fatigue is an important symptom that can interfere in the transition to motherhood (*McQueen & Mander, 2003*). During pregnancy, fatigue is associated with depression, anxiety, fear and duration of labour (*Fairbrother et al., 2008*; *Hall et al., 2009*), and prematurity (*Ko, Wu & Chang, 2002*). During the postpartum period, fatigue has been linked to depression (*Wade, Giallo & Cooklin, 2012*), sleep alterations (*Hung & Chen, 2014*), breastfeeding difficulties (*Wambach, 1998*), and problems bonding with the baby (*Lai et al., 2014*). In early parenting, it has been found that higher parental fatigue is associated with lower parental sense of confidence, higher parenting stress and less optimal parenting behaviours (*Cooklin, Giallo & Rose, 2012*).

Fatigue is a symptom linked to the process of continual adaptation that begins during pregnancy and continues through the transition to motherhood (*Runquist, 2007*). Various instruments have been used previously during pregnancy and postpartum to measure fatigue, depending on whether the aim was to obtain broad and detailed information about the experience of fatigue, using multidimensional scales, or to measure the intensity or severity of fatigue, using unidimensional scales (*Dittner, Wessely & Brown, 2004*; *Whitehead, 2009*). The Fatigue Assessment Scale (FAS) (*Michielsen, De Vries & Van Heck, 2003*) is a tool that provides information about the physical and psychological aspects of fatigue, and provides a single overall score measuring its intensity. The FAS offers adequate characteristics as an instrument that measures patient reported outcomes (*US Department of Health and Human Services FDA Center for Drug Evaluation and Research et. al., 2006*), and this is because it is brief, it does not overburden the respondent, it has the capacity to discriminate between cases and non cases in clinical groups, it is sensitive to change when evaluating interventions, and it has good psychometric properties.

The FAS is a self-reported questionnaire consisting of 10 items with a 5-point Likert response scale ranging from "1 = never to "5 = Always". Five items reflect the physical component and five the psychological component.

Of these items, eight reflects negative aspects on fatigue states (1, 2, 3, 5, 6, 7, 8, 9). The remaining two items (#4 and #10) are worded inversely and are given reverted scores, indicating energy states. The FAS has been used in the context of postpartum and early parenthood among mothers (*Giallo, Wade & Kienhuis, 2014*) and fathers (*Mellor & St. John, 2012*; *Seymour et al., 2014*).

The Cronbach's alpha coefficient of the scale is above 0.80. The factor structure of the FAS has been considered unidimensional (*Michielsen et al., 2004*). However, some studies have described problems with two of the items, #4 and #10 (*Kalkanis, Yucel & Judson, 2013*; *Giallo, Wade & Kienhuis, 2014*), which negatively affect the internal consistency of the scale. This is probably caused by the negative wording (*Giallo, Wade & Kienhuis, 2014*). Furthermore, the suitability of the content of item #10, pertaining to problems of concentrating, is under debate, owing to the possibility that this item is measuring depression (*De Vries, Van der Steeg & Roukema, 2010*).

Given the importance of fatigue as a symptom during pregnancy and postpartum, various international (*World Health Organization, 2013*) and national (*Ministerio de Sanidad, Servicios Sociales e Igualdad, 2014*) guidelines recommend it should be studied in motherhood in general, and particularly during the postpartum period. However, fatigue has been studied very little in the Spanish population, especially in the context of motherhood. There are no publications regarding a Spanish version of the FAS. Hence, this study aims to: (1) validate the FAS at a linguistic level, (2) test its factorial structure, and (3) provide evidence of convergent validity, discriminant validity, as well as discriminative validity (comparisons between known groups).

## MATERIALS & METHODS

### Design

An instrumental design (*Dios & Melendez, 2005*) was conducted, including a two-phase process. First, the scale was translated into Spanish and the content validity was examined, and second, testing was conducted to establish reliability and validity.

### Participants

A convenience sample of 870 women was recruited from 17 regional hospitals in Eastern Spain between October 2013 and June 2014 during their in-hospital postpartum stay. All participants were able to read and write in Spanish. Multiple pregnancies along with severe health problems in the new-born, and processes that required the new-born to be taken into a neonatal unit during the time the mother was hospitalized were excluded.

### Translation and cultural validity

Permission from the authors of the FAS was obtained in order to use and translate it into Spanish. The translation was developed by forward–backward translations from the original English version (*Muñiz, Elosua & Hambleton, 2013*). There were two independent translations, developed by two native Spanish-speaking translators fluent in English. Translators were asked to grade the difficulty of the translation using a numeric rating scale from 1 (not at all difficulty) to 10 (maximum difficulty) and describe the changes they made to each item during translation in order to maintain the semantic equivalence. Both versions were compared by a panel of four experts consisting of two midwives and two psychologists and, after having reached a consensus, the instrument was prepared in a single document. This consensus version was then translated back to English, by two native English-speaking translators fluent in Spanish, without prior knowledge of the original version. This was then compared with the original, to ensure conceptual and semantic equivalence. The expert panel agreed that the Spanish version demonstrated semantic and grammatical equivalence, and one of the original English authors confirmed this aspect. Cognitive interviews of the Spanish version of the FAS was performed with ten pregnant and ten puerperal women to evaluate the adequacy of the format and presentation as well as to assess the clarity and interpretation of each item and response option.

## Data collection

Participants completed a battery of self-administered questionnaires on the day of their hospital discharge. The battery included, in addition to the FAS-e, a questionnaire on sociodemographic variables and items generated ad hoc, using an analogical visual 100-point scale to obtain information about: (1) perceived pain related to their labour or caesarean in the last 24 h and, (2) difficulty breastfeeding during their stay in hospital. The level of maternal self-efficacy to breastfeed was also measured by means of the short form of the Breast Feeding Self Efficacy Scale (BSES-SF) (*Oliver-Roig et al., 2012*). Data on obstetric variables at discharge were also collected from clinical records.

## Data analysis

We calculated the mean and standard deviation of the item scores and, in order to assess floor and ceiling effects, the proportion of respondents with the lowest or highest possible score were described. We calculated missing data from each item to assess acceptability.

To examine the factorial structure of the scale we included participants who responded to all items of the FAS scale ($n = 803$). Also, as a cross-validation strategy, we randomly divided the sample into two sub-samples of approximately equal size, one for development and another for validation. We used exploratory factor analysis (EFA) and confirmatory factor analysis (CFA) with the first sample. The CFA models examined were based on previous findings on the FAS scale and on the EFA and CFA results themselves (examination of the modification indexes, the error terms correlations, and factor loadings). According to the FAS scale background, we examined models of a single factor (the model with the highest theoretical and empirical support), two correlated factors (physical fatigue and mental fatigue), and variations of these models allowing correlated errors (two items of the scale had a positive wording, i.e., positive states of fatigue, and the rest, eight items, a negative wording, i.e., negative states of fatigue). In addition to these more traditional models, we used bifactorial models. These models stipulate a general factor and one or several specific factors, constrained as not correlated with each other, allowing to identify if the general factor explains the greater part of the variance in spite of the specific factors (*Deng, Guyer & Ware, 2015*). These models have been specifically recommended to examine whether a measure is sufficiently one-dimensional in the presence of secondary dimensions (i.e., can it be assessed as an essentially unidimensional measure or not, if secondary dimensions such as physical and mental subdomains of a general measure of fatigue, or if the effects of wording of items distort the interpretation of the scale). With the second sample, we examined the robustness of the CFA models analysed with the first sample. Due to the ordinal nature of the items and the non-normality of their distributions we used methods suitable for categorical data, i.e., polychoric correlation matrices, DWLS estimator in CFA and minimum residual method (MinRes) in EFA.

As indexes of goodness of fit of the models we use: the comparative fit index (CFI); goodness of fit index (GFI), the root mean square error of approximation (RMSEA), and the standardized root mean square residual (SRMR). Guidelines suggest that models with CFI and GFI close to 0.95 or higher, RMSEA close to 0.06 or lower, and SRMR close to 0.08 or lower represent good-fitting models (*Hu & Bentler, 1999*).

Reliability of the translated FAS-e was assessed using the following criteria: Cronbach's alpha coefficient, adjusted item-total correlation, and estimation of $\alpha$ when an item was removed from the scale.

Further, evidence of discriminant validity and relationships with other variables according to previous evidence was obtained. The total scores from the FAS Scale were expected to be higher for mothers having undergone a caesarean section (*Cheng et al., 2014*; *Kilic et al., 2015*; *Lai et al., 2014*; *Rowlands & Redshaw, 2012*) and experienced increased pain in the postpartum period (*Fishbain et al., 2003*; *Wambach, 1998*). It was also expected that women who experienced breastfeeding problems or who had lower levels of self-efficacy in breastfeeding would obtain higher scores on the FAS scale, given that breastfeeding is one of the main care demands of the postpartum period (*Dennis, 1999*; *Wambach, 1998*).

Scores on the FAS scale were also evaluated for other sociodemographic types of variables, such as age, or obstetric variables object of controversy in previous studies with regard to their relationship with fatigue. The Kolmogorov–Smirnov test was used to compare the FAS-e score distribution with the normal distribution of variables. The Student's *T* test and Pearson correlations were used to compare hypotheses. The LISREL software (*Jöreskog & Sörbom, 2006*) was used for all these factorial analyses. Other analyses were conducted using SPSS 22.0 (*IBM Corp., 2013*).

## Ethical aspects

The women in the sample were approached by a research assistant and informed about the study. When the women expressed an interest in participating in the study, a screening interview was performed and, if eligible, written consent to participate was obtained. Approval was received from the research ethics committee of the Directorate General of Public Health and Public Health Research Center ("*Dirección General de Salud Pública y Centro Superior de Investigación en Salud Pública*") of the Valencian Community (Spain).

## RESULTS

### Semantic equivalence

None of the 10 items on the scale were considered inappropriate for the Spanish context. It was not necessary to make any changes to any of the 10 items. According to the translators' evaluation, the average difficulty of the translation by item was 1.8 ($SD = 0.4$, range 1.25 to 2.50), and did not rise above 4 for any item on the established scale of 1–10.

Item #1 "I am bothered by fatigue" sparked debate among the translators owing to the existence of two possible terms for the translation of bothered, "*molesta*" or "*preocupada*". The "*preocupada*" option was chosen, since "*molesta*" was considered to be more closely linked to discomfort or pain rather than fatigue.

Reading and comprehension problems were not encountered in cognitive interviews, and the average time for completion was 85.4 s ($SD = 25.3$, range 50–125).

### Description of sample

The average age of the sample was 32.4 years old ($SD = 4.9$, range 15–46). Table 1 provides the other characteristics of the sample as well as percentiles of the FAS-e scores.

**Table 1  Characteristics of the study sample ($n = 870$).**

| Variable | $n$ (%) |
|---|---|
| Country | |
| Spain | 751 (86.5) |
| Other countries | 117 (13.5) |
| Education level | |
| Primary school | 378 (44.3) |
| High school/University degree | 476 (55.7) |
| Marital status | |
| De facto or married | 702 (82.6) |
| Single | 148 (17.4) |
| Living with partner | |
| Yes | 817 (95.9) |
| No | 35 (4.1) |
| Income | |
| <12,000 € | 280 (35.6) |
| >12,000 € | 506 (64.4) |
| Parity | |
| Primiparae | 428 (49.2) |
| Multiparae | 442 (50.8) |
| Mode of delivery | |
| Vaginal delivery | 692 (79.6) |
| Caesarean delivery | 177 (20.4) |
| Breastfeeding initiation | |
| Yes | 770 (88.7) |
| No | 98 (11.3) |
| Percentiles of the FAS-e scores | Score |
| 10 | 13 |
| 20 | 14 |
| 25 | 15 |
| 30 | 16 |
| 40 | 17 |
| 50 | 19 |
| 60 | 20 |
| 70 | 21 |
| 75 | 22 |
| 80 | 23 |
| 90 | 26 |

**Table 2** FAS-e items reliability results, floor and ceiling effects, means (*M*) and standard deviation (*SD*).

| Item | *n* | Corrected item-total correlaction | Crombach's Alpha if item deleted | Floor ("1" %) | Ceiling ("5" %) | *M* | *SD* |
|------|-----|------|------|------|------|------|------|
| 1. I am worried about fatigue. | 852 | 0.425 | 0.79 | 42.0 | 1.3 | 1.76 | 0.81 |
| 2. I get tired very quickly. | 857 | 0.593 | 0.78 | 36.9 | 1.1 | 1.85 | 0.85 |
| 3. I don't do much during the day. | 827 | 0.411 | 0.80 | 40.9 | 2.9 | 1.94 | 1.01 |
| 4. I have enough energy for everyday life.[a] | 854 | 0.415 | 0.80 | 22.8 | 5.2 | 2.72 | 1.21 |
| 5. Physically, I feel exhausted. | 852 | 0.534 | 0.78 | 22.2 | 2.1 | 2.05 | 0.78 |
| 6. I have problems starting things. | 847 | 0.572 | 0.78 | 54.5 | 1.1 | 1.59 | 0.78 |
| 7. I have problems thinking clearly. | 849 | 0.571 | 0.78 | 64.2 | 0.9 | 1.47 | 0.74 |
| 8. I feel no desire to do anything. | 847 | 0.603 | 0.78 | 51.2 | 0.9 | 1.61 | 0.76 |
| 9. Mentally, I feel exhausted. | 849 | 0.612 | 0.77 | 49.7 | 0.9 | 1.68 | 0.84 |
| 10. When I am doing something, I can concentrate quite well.[a] | 849 | 0.316 | 0.82 | 29.1 | 6.4 | 2.61 | 1.30 |

Notes.

[a] Items #4 and #10 have been reverted before carrying all computations.

The questionnaire response rate was good. The percentage of data lost did not exceed 5% for any of the items on the scale. The mean total score obtained on the FAS-e score was 19.17 with a standard deviation of 5.62 and the proportion of responses with the lowest possible score, "1", was above 15% for all items (Table 2).

## Factor analysis results, first subsample (*n* = 406)

The EFA analysis on the first sub-sample (*n* = 406) identified three possible factor solutions: unifactorial, two factors and three factors (the first six components had the following eigenvalues: 4.72, 1.17, 0.83, 0.78, 0.61, 0.54). In the unifactorial solution, the loadings of the items ranged between 0.34 and 0.81. The correlated two-factor solution (Promax rotation, *r* = 0.46) brought together the eight items with negative wording in one factor and the two items with positive wording in the other, with a clear simple structure. The three-factor solution brought together four items of physical fatigue in one factor, four items of mental fatigue in a second factor and the two items with positive wording in the third factor. The four-factor solution was uninterpretable and clearly meant an overfactorization.

Table 3 describes the CFA models examined with the two sub-samples, as well as the results of the fit indexes of the models. In general, the models examined were those stipulated a priori, since the results of EFA and CFA did not provide any notable novelty (i.e., there was no indication to estimate more error terms correlations between items than that stipulated a priori between error terms of the two items with positive wording).

Regarding the CFA results, the one-dimension model (Model 1a) fitted well to the data, with loadings ranging from 0.35 to 0.83; its fit improved slightly (Model 1b) when the correlation between the error terms of items with positive wording ($\theta_{4,10} = 0.34$) was allowed. The correlated two-factor model presented only slightly better indices than the unidimensional model and estimated a very high correlation between both factors

**Table 3  Confirmatory factor analyses of the Fatigue Assessment Scale: overall model fit.**

| Model | $\chi^2_{S-B}$ | df | CFI | GFI | SRMR | RMSEA |
|---|---|---|---|---|---|---|
| Subsample 1, $N = 406$ | | | | | | |
| Model 1a | 108.8 | 35 | 0.98 | 0.99 | 0.064 | 0.072 |
| Model 1b | 83.8 | 34 | 0.98 | 0.99 | 0.048 | 0.060 |
| Model 2a | 87.5 | 34 | 0.98 | 0.99 | 0.059 | 0.062 |
| Model 2b | 61.44 | 33 | 0.99 | 0.99 | 0.040 | 0.046 |
| Model 3a | 84.99 | 33 | 0.98 | 0.99 | 0.048 | 0.062 |
| Model 3b | 50.94 | 25 | 0.99 | 1.00 | 0.035 | 0.051 |
| Model 4a | – | – | – | – | – | – |
| Model 4b | 38.5 | 24 | 1.00 | 1.00 | 0.032 | 0.039 |
| Subsample 2, $N = 397$ | | | | | | |
| Model 1a | 119.1 | 35 | 0.97 | 0.98 | 0.064 | 0.078 |
| Model 1b | 114.1 | 34 | 0.97 | 0.99 | 0.064 | 0.077 |
| Model 2a | 73.0 | 34 | 0.99 | 0.99 | 0.052 | 0.054 |
| Model 2b | 62.4 | 33 | 0.99 | 0.99 | 0.045 | 0.047 |
| Model 3a | 112.7 | 33 | 0.97 | 0.99 | 0.060 | 0.078 |
| Model 3b | 50.94 | 25 | 0.99 | 1.00 | 0.035 | 0.051 |
| Model 4a | – | – | – | – | – | – |
| Model 4b | 41.63 | 24 | 0.99 | 0.99 | 0.033 | 0.043 |

**Notes.**

$\chi^2_{S-B}$, Chisquare Satorra-Bentler; CFI, Comparative Fit Index; GFI, Goodness of Fit Index; RMSEA, Root Mean Square Error of Approximation; SRMR, Standarized Root Mean Square Residual; Model 1a, one-dimensional model; Model 1b, model 1a with correlated errors between items with positive wording, items #4 and #10; Model 2a, two correlated factors: physical fatigue (five items) and mental fatigue (five items); Model 2b, model 2a with correlated errors between items #4 and #10; Model 3a, bifactorial model of a general factor and a specific factor with items with positive wording; Model 3b, bifactorial model of a general factor and two specific factors, one with items with positive wording and the other with items with negative wording; Model 4a, bifactorial model of a general factor and two specific factors, one for the items of physical fatigue and the other for those of mental fatigue. It did not converge in any of the two subsamples; Model 4b, model 4a with correlated errors between items #4 and #10.

($r = 0.88$). As with the unidimensional model, allowing the correlation between the errors of the two items with positive wording (Model 2b) enabled slightly improving the fit.

To what extent did the multidimensionality identified in these models, i.e., the effects of wording and subdomains of mental and physical fatigue obscure a unidimensional interpretation of the FAS scale? First, according to the results of the bifactorial models that included wording effects (models 3a and 3b), although there was a secondary factor of positive wording (Model 3a), its existence hardly had an influence on the loadings of these items in the general factor (their loadings were approximately the same as those of the pure unidimensional model, Model 1a). Also, the model including a specific factor composed of the eight items with negative wording (Model 3b) showed the irrelevance of this specific factor (loadings not significant or less than 0.2, except for one item, in the negative wording specific factor and loadings in the general factor very close to those of the unidimensional model). Therefore, the wording of the items did not interfere with the unidimensional interpretation of the scale. Secondly, regarding the influence of mental and physical fatigue subdomains, Model 4a did not converge but Model 4b, including the specific factors of mental fatigue and physical fatigue in addition to the estimation of

the correlated error terms between items #4 and #10, did converge. This model achieved the best fit indexes (it incorporated all sources of variation found: general fatigue factor, positive wording of two items and fatigue subdomains) and showed a strong general factor (loadings were slightly lower than those of the unidimensional model with a difference equal to or less than 0.05 in nine items and the largest difference was 0.08 in item 8), two weak specific factors (only one item in each specific factor had loadings greater than 0.3 and an error terms correlation between items #4 and #10 of 0.33). In summary, the FAS scale was sufficiently unidimensional, despite the presence of method effects (positive wording) and effects related to the subdomains of fatigue.

### Factor analysis results, second subsample ($n = 397$)

The results on the second (validation) sub-sample ($n = 397$) corroborated those of sub-sample 1, however in this case the effects of method were somewhat smaller and those of subdomains somewhat larger (i.e., the correlation between physical and mental fatigue was 0.81, compared to 0.87 in the first sub-sample).

### Internal consistency reliability

Cronbach's alpha was .80. FAS-e items reliability results, means and standard deviations are provided in Table 2. The alpha value decreases if any item is eliminated, with the exception of eliminating item #10, which increases it to .82. The item-total correlations for the 10 items were all >.30, positive and significant in this study, and ranged between .32 and .61.

### Discriminant validity, comparisons between known groups and correlations with other variables

Total score average of the FAS-e was 19.17 ($SD = 5.62$), showing a non normal distribution ($z = 2.46$, $p = .01$). The average FAS-e score was 18.92 ($SD = 5.4$) for women with vaginal delivery and 20.17 ($SD = 6.15$) for women with caesarean, reflecting significant differences between the FAS-e scores ($t = -2.53$; $p < 0.05$). The *Pearson* correlation coefficient between pain intensity during postpartum (Likert 0–100) and total FAS-e score was .18; $p < .01$. The *Pearson* correlation coefficient between breastfeeding difficulties (Likert 0–100) and FAS-e score was .12; $p < .01$. The *Pearson* correlation coefficient between BSES-SF and FAS-e score was $-.25$; $p < .01$. No differences were found related to age ($r = -0.04$; $p = .24$), parity ($t = 87$; $p = .38$) and type of feeding at discharge ($t = -1.75$; $p = .08$).

## DISCUSSION

In this study, we developed the Spanish version of the FAS, provided evidence of validity and reliability, and analysed its factorial structure, considering physical and mental subdomains and the effect of the wording of items, in a sample of postpartum women at hospital discharge.

The translation process was systematic and rigorously conducted to ensure that equivalence was established. Only one item (#1) sparked debate in translation owing to the dual possible translation of "bothered" as either "*molesta*" or "*preocupada*". Once consensus was reached regarding its translation, it did not present any problems in the pilot test or the definitive sample.

The scale presented a floor effect for all items. This greater proportion of responses at lower levels of fatigue in the sample of this study could be due to the fact that the participants were young healthy women. Other possible causes for the lower level of fatigue among the sample studied are that the data were obtained in the morning, when levels of fatigue are lower (*Elek, Hudson & Fleck, 2002*), and on the date of discharge, so participants might have had a tendency to report less fatigue, influenced by their imminent return home with their baby (*Fredrickson, 2000*; *Tzeng et al., 2008*). No evidence was provided on the floor effect in the scoring of items in previous studies that used the FAS scale on a sample of postpartum women.

The findings on the dimensionality of the FAS scale indicate that it was sufficiently unidimensional, despite the presence of a certain multidimensionality derived from the positive wording effects of two of the items, in contrast to the negative wording of the rest of the eight items, and despite the effects of the subdomains of the scale, i.e., physical fatigue and mental fatigue. We modelled these effects in two ways: (i) through factor models with correlated errors (to estimate the effects of item wording) as well as correlated two-factor models (to estimate the effects of fatigue subdomains), and (ii) bifactor models that incorporated both types of effects as specific factors, in addition to a general factor. The findings of the bifactorial models revealed that these sources of variation were secondary and in no way distorted the interpretation of the scale as a solid one-dimensional measure. Unidimensionality of the trait is a general finding in the literature (*Michielsen, De Vries & Van Heck, 2003*; *Michielsen et al., 2004*) as well as a specific finding when studied in a population similar to ours, i.e., postpartum women (*Fairbrother et al., 2008*; *Tsai et al., 2014*). In addition, our findings are also compatible with the content specifications of the scale, since the two subdomains, mental and physical fatigue, underlie but do not distort the unidimensionality of the scale. And finally, our findings corroborate the effects that introduce the contrast of the positive wording of two items versus the negative wording of the rest of the items on the scale (*Kalkanis, Yucel & Judson, 2013*). Although we performed a methodological interpretation of the wording effects of the items, a more substantive interpretation is also plausible (*Deng, Guyer & Ware, 2015*): items with negative wording would be indicators of fatigue, whereas items with positive wording would be indicators of energy (and both would be indicators of the same construct: vitality).

The FAS-e offers acceptable internal consistency. Cronbach's alpha for the FAS-e (.81) was lower than the original version (.90), but higher than the minimum acceptable level of .70, and similar to the value obtained in other populations/samples (*De Vries et al., 2004*; *Michielsen, De Vries & Van Heck, 2003*).

In this study, as in others (*Lai et al., 2014*; *Cheng et al., 2014*; *Kilic et al., 2015*; *Rowlands & Redshaw, 2012*), women who had undergone a caesarean showed higher fatigue than those who had vaginal deliveries. Additionally, as described previously, women with a higher level of pain during the postpartum period (*Fishbain et al., 2003*; *Wambach, 1998*) or higher perceived difficulty with breastfeeding (*Wambach, 1998*) were more fatigued. Furthermore, the data confirmed, as previously suggested (*Dennis, 1999*), that fatigue was one of the symptoms relating to self-efficacy in breastfeeding.

Increased age has been associated with greater obstetric problems and, therefore, high levels of fatigue postpartum (*Taylor & Johnson, 2013*). First-time mothers can present greater fatigue related with longer labours and inexperience in parenting (*Tzeng et al., 2008*; *Taylor & Johnson, 2013*). Our data did not suggest differences in levels of fatigue relating to these variables. The fact that the data were obtained prior to postpartum discharge and before the women had had to cope with the challenges of daily life and the re-establishment of family roles, could explain the lack of differences relating to the parity variable. As for the influence of the mother's age, another recent study similarly found no significant differences in levels of postpartum fatigue in primaparous women according to age (*Tsuchiya et al., 2015*).

In the future, it would be desirable to test the performance of the instrument at different points during the postpartum period, or following interventions, to ascertain the instrument's sensitivity to change in this population group. It would also be interesting to explore how the instrument performs for fathers, who are also affected by childbirth.

## CONCLUSIONS

The Spanish version of the FAS (FAS-e) has been obtained and its preliminary validation has been established. The FAS-e is a sufficiently unidimensional, valid, and reliable tool to study fatigue in postpartum women in Spain.

### Funding

This work was funded by the General Sub-Directorate for Evaluation and Promotion of Research (Institute of Health Carlos III, ISCIII) and co-funded by the European Regional Development Fund (FEDER) ''A way to make Europe'' (Project reference PI14/01549). The funders had no part in designing the study, collecting, analyzing, interpreting or influencing the writing of the data or the decision to publish.

### Grant Disclosures

The following grant information was disclosed by the authors:
General Sub-Directorate for Evaluation and Promotion of Research (Institute of Health Carlos III, ISCIII).
European Regional Development Fund (FEDER): PI14/01549.

### Competing Interests

The authors declare there are no competing interests.

### Author Contributions

- Antoni Cano-Climent conceived and designed the experiments, performed the experiments, contributed reagents/materials/analysis tools, wrote the paper, prepared figures and/or tables, reviewed drafts of the paper.
- Antonio Oliver-Roig and Miguel Richart-Martínez conceived and designed the experiments, performed the experiments, analyzed the data, contributed reagents/materials/analysis tools, wrote the paper, prepared figures and/or tables, reviewed drafts of the paper.
- Julio Cabrero-García analyzed the data, contributed reagents/materials/analysis tools, wrote the paper, prepared figures and/or tables, reviewed drafts of the paper.
- Jolanda de Vries reviewed drafts of the paper.

## Ethics

The following information was supplied relating to ethical approvals (i.e., approving body and any reference numbers):

Ethical approval was received from the Research Ethics Committee of the Directorate General of Public Health and Public Health Research Center ("Dirección General de Salud Pública y Centro Superior de Investigación en Salud Pública") of the Valencian Community (Spain).

## Data Availability

The raw data has been included as a Supplemental File.

## Supplemental Information

Supplemental information for this article can be found online at http://dx.doi.org/10.7717/peerj.3832#supplemental-information.

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
