# Peer review of "The Spanish version of the Fatigue Assessment Scale: reliability and validity assessment in postpartum women"

_PeerJ, doi:10.7717/peerj.3832_

## Round 0.1 · original submission · Major Revisions

· Academic Editor

Major Revisions

We thank you for your submission to PeerJ. The three reviewers are positive about the strengths of the manuscript, but they have also identified a number of amendments which should be addressed prior to resubmission.

Of particular note, the English needs improvement.

I would therefore recommend you get additional assistance on this aspect of the manuscript. We suggest you have a fluent, preferably native, English-language speaker thoroughly copyedit your manuscript for language usage, spelling, and grammar. If you do not know anyone who can do this, you may wish to consider employing a professional scientific editing service.

Thank you for considering PeerJ for publication of your manuscript.

Best Wishes,

Reviewer 1 ·

Basic reporting

An article whose findings are important to those with closely
related research interests. This manuscript present and discusses the validation of a scale to assess fatigue during the puerperium period. As author explain in the discussion section, this may be an apropiatte scale to be used in different time points besides the early postpartum period and helpful to assess fatigue evolution along all the postpartum

As english is not my (reviewer) mother language , I would suggest an expert english language review.

Experimental design

Reserch question is well designed and methodology used is adequate. Methods are well described and easy to understand.

Hipotesis stated by authors is fulfilled.

Validity of the findings

Analisys is well performed and is confirmatory. However as a suggestion to authors I would reccomend to obviate the exploratory analisis on item #4 and #10. It is already demonstrated and confirmed the saturation of these two items, and this exploratory analys may be considered as redundant.

Comments for the author

Manuscript is well written and easy to understand.
Exploratory analysis for items #4 and #10 may be considered as redundant. Please consider reviewing
This scale may be a good instrument to assess fatigue during postpartum period in Spain, and useful for international comparisons
Minor comments:
Line 122: authors use the acronym BSES-SF for the first time, please explain
Line 135: it seems a wording mistake " closet o 0,95" , as i understand it is intended to be " close to 0,95". please confirm or correct

·

Basic reporting

The authors have developed a clear and rigorous scientific paper, with clear development and adequate language. The introduction is complete and allows the reader to be correctly placed in the context of the research. The references cited are useful and up-to-date.

Complete graphic resources are provided and raw data is supplied.

I suggest you take note of the following points about your figures and tables:

Figure 1 - Write "Spanish" (as proper noun).
Table 1 - Write "Table 1" (in English) instead of "Tabla 1" (Spanish).
Table 2 - Add "#" symbol at ítem 4 and ítem 10. They can also do the same in the lines 142, 201, 210 and 217.
Table 3 - Review spelling: "standard deviation".

Experimental design

It is an original investigation with a clear purpose, which respects ethical codes, and whose method is simple and easily applicable.

Validity of the findings

The conclusions are clearly related to the research approach. The validity of the results will allow an application of the scale in the daily consultations.

Comments for the author

I suggest you take note of the following format and spelling points:

Abstract - Review spaces between numbers, words and fulls stops.
Line 33 - Close quotation marks at the end of the sentence,
Line 40 & 41 - I suggest you to link the expresions "fear and duration of labour" in one sentence.
Line 74, 101 & 104 - Review spaces between words and brackets.
Line 105 - Review spelling "maintain".
Line 110 - Delete accent "version". Delete dead space at the end of the sentence.
Line 111 & 112 - Review agreement between singular/plural in these sentences.
Line 132 - Review spelling "and" ; "comparative".
Line 145 - Review full stop of the sentence.
Line 158 - Tabulate the beginning of the sentences.
Line 159 - Delete innecesary full stop in the middle of the sentence.
Line 162 & 164 - Review dead spaces.
Line 179 - Write "molesta" and "preocupada" in italics (Spanish words)
Line 203 - Full stop at the end of the sentence required.
Line 216 - To maintain uniformity, I suggest you to to write "table 3" according to the other styles you use; "Table 3".
Line 244 - Write "molesta" and "preocupada" instead of "molesto" and "preocupado" (maintain uniformity).
Line 253 - Review space between full stops and words.
Line 270 & 271 - Write "Model 1" and "Model 2" instead of "model 1" and "model 2" (maintain uniformity).

·

Basic reporting

The paper “The Spanish version of the Fatigue Assessment Scale: Reliability and validity assessment in postpartum women (#17227)” presents the cultural validation and initial psychometric properties of the FAS in postpartum women. The structure of the paper is adequate, contents are properly arranged in their sections, comments are not mixed with results, the discussion is well supported by the reported results, the methodology used is ade4quate for the aim of the study and the sample used has enough size.
The paper is written in a way easy to understand although some English editing would be welcome. Some of the terms and expressions difficult to understand are pointed out bellow.

Experimental design

The design is adequate for the aims of the study although some data analysis should be improved.

Validity of the findings

The validity of results is supported by the large sample used. Methods are adequate, but perhaps incomplete, and regular procedures for questionnaire cultural adaptation have been followed. Sensitivity to change is still to be tested.

Comments for the author

To main issues are to be distinguished in the paper. One is the cultural adaptation into Spanish of the original FAS, the other is the existing differences in structure between versions.
As for the first part, the paper is straightforward and results are really worth, specially taking into account the large sample used in the validation process. These findings are very useful in usual clinical practice and it is interesting to have a Spanish version of the instrument. Perhaps it would be interesting to have some additional data about the normative values obtained with this the sample, such as percentiles, in order to make it easier to know the meaning of the fatigue value obtained by a patient.
As for the second part, there is nothing wrong in finding that the structure of an instrument might be different in other cultures. Nevertheless, I personally feel that the approach followed by the researchers is incomplete, but easy to work out. Reading the two items giving some trouble, they do remind other items used to measure anxiety and depression (like in the HADS scale). It is not easy to select proper items for a scale and the researchers are not the ones to be blamed for it. But some other methodological strategies are available to ensure that this two items are rather “noisy” different from the ones used by the authors.
First. Estimated factor loadings are rather mild for a uni-dimensional scale. It could be the case that the authors have not considered the ordinal nature of Likert response scales and that the estimations method is not really the best one. I would recommend using MPlus defining ordinal metric for itmes and a better estimation method such as MLMV or WLSMV. Lisrel using the polychoric correlation matrix could also give good results. It would be interesting to mention in the paper the structure obtained by the original authors in English to compare.
Second. The extraction method used in the exploratory factor analysis should be reported, and the obtained structure matrix to (even if the graph of rotated factor loadings is reported). Furthermore, it is not a good idea to use a Varimax (orthogonal) rotation when you are expecting items to be load in a single dimension. An oblique rotation would be more suitable, and the correlation between factors will give a hint on the real proximity of items in the rotated structure. The authors should keep in mind that they are not willing to improve the scale and hence they should not discard any item. At most, they should find better versions of the original items (¡and repeat the data gathering!).
Third. The varimax solution will not give you the proper picture of the differential functioning of the two conflictive items. It is known that bi-polar dimensions (with a positive and negative end, and with items located at each one of the ends) tend to be treated as separated or independent dimensions by the rotation algorithms. The problem here is that the rotation procedure is not able to distinguish dimensions at 180º from those at 90º. This is a known analytical artifact, which should be thoroughly checked.
Fourth. Given that the authors think that the negative wording of the offending items could be introducing a methodological bias they could use more sophisticated structures to test if this is so. For instance, setting free the correlation between error terms for items 4 and 10 (theta-10-4) would estimate the amount of relation of this two items due to wording, and subtract it from their relation with the underlying latent dimension. Other procedures as a bi-factor structure, a multy-trait multi-method, or a second order factor structure will also help understanding the structure and assessing the response bias due to negative wording.
Fifth. The authors could always cross-validate the results splitting their sample and comparing both subsamples.

From the results obtained (Line 225) It looks like there is not a big departure from the normal distribution (z=2.46) for the size of the sample used (n>850). If the authors are willing to compare mean values, they could safely use the t-test, since the distribution of possible sample mean values will be normal with such group sizes (no matter if the distribution of scores are not). Furthermore, it is more interesting to adjust for unequal group variabilities (when selecting the proper degrees of freedom for the t-test) than to prevent for deviations from normality (using the Man-Whitney U test). Z scores have the benefit that they give a feeling of the effect size for differences between groups, something much more difficult to do using the U.

Some other minor comment follow.
Line 58. The way the sentence is written makes you think that the fact that it is sort will make it more discriminant, or will give better psychometric properties. Please separate concepts.
Line 58. Consider changing the expression “since it is brief, does” to “since it is brief, it does”.
Line 62. Consider using “1=Never to 5=Always”.
Line 63. Please modify the sentence. Only two items 4 and 10 are worded in a negative way, the other eight are positively worded but they ask for negative aspects or negative impacts on health i.e.: deterioration.
Line 65. I suspect that the scores given to items 4 and 10 are not “negative scores” but rather “reverted” scores. See also Line 141.
Lines 69. “reliability indices”. Only one reliability coefficient is reported. It should be singular.
Line 105. Use “maintain”.
Lines 109-110. The team ensuring semantic equivalence should have been the original English authors and not the developing team.
Line 111. ¿How many interviews where carried out?
Line 132: Use “and” for “ant”.
Line 168. The name of local institutions is never translated. Although you may offer a translation in brackets.
Line 182. How many cognitive interviews were carried out?
It is customary to report a dispersion statistic (e.g.: standard deviation, interquartile range, etc.) accompanying the mean when average values are reported. See lines 176, 183 & 193.
Line 195, 216. When particular figures and tables are addresses in the text, they should be mentioned in capital letters, i.e.: “Table 3”.
Lines 199, 201, 259, 270. Please use the term “Goodness-of-fit statistics” or “Goodness-of-fit indices”, instead of “Indices of fit”, which is the standard for this kind of indices. Otherwise, use “fit indexes”.
Consider rephrasing Line 199: “Goodness-of-fit statistics for the single dimension model containing all items (Model 1) were good.”
Page 202. It is more frequent to use “factor loading” instead of “factor weight”, although it is not strictly incorrect.
Line 208, please use chi-square (or 2) instead of c2. Furthermore, report the degrees of freedom after the chi-square value and using an equal sign: chi-square=2242.8, df=45, p<0.01.
Line 229, 230. Use “FAS-e score” instead of “Score FAS-e”.
Line 231. Use an equal sign between Spearman Rho and the reported value.
Line 240. “owing to variance in method”. Consider using “due to a methodological bias”.
Line 254. Consider using “no evidences have been provided” instead of “have data been provided”.
Line 257. Consider using “support” for “attest to”.
Line 265: it is difficult what the authors mean by “without leaving the common dimension”.
Line 266. “problem of method variance”. Consider using “problem of variability due to method” or “variability due to method bias”. And modify Line 279 accordingly.
Line 278: consider using “increase” instead of “exceed”.
Line 281: Please review the following wording: “a sample of mothers to children aged 0 to 5 years old”.
Line 290: I am not sure what the authors mean when using “diminish”. Perhaps they are intending to mean “discard”, but they should note that the kind of design they are using is not prospective and hence they may not infer if the FAS scoring (after birth) has any screening properties on self-efficacy (as related to vaginal delivery).

Table 3. It should be mentioned that items 4 and 10 have been reverted before carrying all computations. Otherwise some results, such as the item-total correlation or the sign of the factor loading are difficult to understand.
Table 3. Review translation for item 1.
Table 3. Items should be numbered to make it easier the reference.

---

## Round 0.2 · Minor Revisions

· Academic Editor

Minor Revisions

The reviewer(s) have recommended publication, but also suggest some remaining minor revisions to your manuscript. Therefore, I invite you to respond to the reviewer(s)' comments and revise your manuscript.

Reviewer 1 ·

Basic reporting

Authors have succsesfully reviewed and modified the draft accordint to comments and suggestions made in previpous review. Paper may be accepted now.
No more comments to add

Experimental design

No more comments to add

Validity of the findings

No more comments to add

Comments for the author

No more comments to add

·

Basic reporting

The new version of the original paper “The Spanish version of the Fatigue Assessment Scale: Reliability and validity assessment in postpartum women (#17227)” has improved noticeably and the answers given by the authors to the reviewer suggestions are sound and clear. I would like to congratulate the authors for their efforts in improving their work.

I would like to pose one minor suggestion. The term “group factor” is used to distinguish the content specific dimensions in the bi-factor solution from the overall factor/dimension (the later including all items). See lines 142, 229, 230, 239, etc. Although changing “group” by “grouping” could make it easier to understand the text it is usual, in the existing literature, to refer to such factors as “specific” factors as opposed to the “general” overall dimension (see Muthen & Muthen, 2010; Morin, Arens & Marsh, 2000; Reise, 2015). I would suggest using “specific” instead of “group”, especially when two samples have been used in the study, and it may induce readers to think about a multi-group cross-validation analysis using both samples. Please also correct it in Table 3 footnotes.

Line 153 and footnote in Table 2. Use “goodness of fit index (CFI)” instead of “goodness fit index (CFI)”

Line 185 and table 1. Use a dot as decimal placeholder instead of a comma.

Line 213. Use “error terms” instead of “residues”. Avoid using “residuals”, since residuals in SEM models refer to the reproduced covariance matrix based on the model parameter estimates.

Experimental design

No commnent

Validity of the findings

No comment

---

## Round 0.3 · accepted · Accept

· Academic Editor

Accept

We thank you for attending to the reviewers comments on your revised
manuscript and am happy to let you know the paper has now been
accepted for publication.

·

Basic reporting

No comment

Experimental design

No comment

Validity of the findings

No comment

Comments for the author

No comment